# Dietary intake and its associated factors among in-school adolescents in Ghana

Thomas Hormenu[ID]*

Department of Health, Physical Education and Recreation, University of Cape Coast, Cape Coast¸ Ghana

* thormenu@ucc.edu.gh

## Abstract

### Introduction

Early-life nutrition related experiences may fuel the emergence of obesity and type 2 diabetes in adolescence. The adoption of unhealthy dietary practices early in life is an indicator of adverse cardiometabolic health in adulthood. In-school adolescents' dietary practices in Ghana have not been explored extensively despite increasing levels of obesity in adolescents. This study sought to examine dietary practices, socio-demographic disparities and the factors influencing dietary choices among in-school adolescents in Ghana.

### Methods

A school-based, cross-sectional study was conducted in the Central region of Ghana in 2017. Using multistage sampling procedures, a total of 1,311 in-school adolescents were selected for the study. A modified version of the generic Global School Health Survey questionnaire on dietary practices was adapted and used for data collection. Percentage and frequency counts were used to report on the dietary practices, while Chi-square was used to determine socio-demographic variations in the dietary practices. Binary logistic regression was used to compute the influence of socio-demographic characteristics of respondents on the prevalence of healthy dietary behavior among in-school adolescents.

### Results

The study revealed the prevalence of healthy dietary practices (49.9%, $n = 654$) among in-school adolescents in the region. The study also found increased frequency in consumption of soft drinks (93%, $n = 1220$) and toffees/sweets (90%, $n = 1183$) among in-school adolescents. However, low intake of breakfast (57%, $n = 749$) was observed among the adolescents. Significant disparities were observed in relation to gender, age, parental communication, academic performance and geographical location in the dietary practices of in-school adolescents. Furthermore, gender (OR = 1.36, $P = 0.007$), academic performance (OR = 2.19, $P = 0.001$) and geographical location (OR = 1.79, $P = 0.001$) were found to be significantly associated with dietary practices among in-school adolescents in the region.

**Data Availability Statement:** All relevant data are within the paper and its Supporting Information files.

**Funding:** The author received no specific funding for this work.

**Competing interests:** No conflict of interest in the conduct of this research and preparation of this manuscript.

## Conclusions

There was low consumption of fruits and vegetables among adolescents. Fruits and vegetables consumption was associated with gender, academic performance and geographical location, and these may be a reflection that knowledge on healthy food choices and availability are important factors influencing dietary choices among in-school adolescents. School health policy interventions aimed at improving nutritional status among adolescents and enhanced fruit and vegetable consumption in the country should take into account the potential benefit of increasing availability of fruits and vegetables in schools, while reducing access to sweets and soft drinks in the schools and communities.

## Introduction

Early-life nutritional experiences may fuel the emergence of obesity and type 2 diabetes in adolescence. Adolescence is a critical period for the formation of lifelong behaviors. It is a period where nutritional needs and explorations increase due to the increased growth rate and changes in body compositions [1]. The dramatic increase in energy and nutrient requirements in this period coincides with other factors that affect adolescents' food choices and nutrient intake and thus, nutritional status. These factors, including the quest for independence and acceptance by peers, greater time spent at school/playgrounds, and preoccupation with self-image, contribute to erratic and unhealthy eating habits [2]. To help prevent diet-related chronic cardiovascular diseases, studies have posited that healthy eating behaviours, including fruit and vegetable consumption, should be established early in life and maintained during adolescence and adulthood [3].

In a national and population-based survey, it was observed that adolescents often fail to meet dietary recommendations for overall nutritional status and for specific nutrient intakes [4]. According to this study, a large number of adolescents received a higher proportion of energy from fat and/or added sugar and had a lower intake of fruits and vegetable than is recommended. In addition, another study conducted by Buxton [5] on Junior High School (JHS) adolescents' dietary practices and food preferences found that the majority (n = 515, 62.8%) of the adolescents usually skipped breakfast before going to school and preferred snacks (soft drinks and pastries items). Buxton concluded that the majority of students did not have healthy eating patterns and habits, as they usually skipped breakfast and preferred food products with high sugar and fat content as snacks among other dietary habits.

Recent global figures have shown alarming numbers of obese and overweight children and young people. In Ghana, 43% of adult population is either overweight or obese [6] while 46.9% of children are either obese or overweight [7]. Contributory factors include lifestyle changes in the consumption of energy dense foods, proliferation of energy dense food near living vicinities of adolescents, and lack of complementary physical activities in the schools and communities. Trends over time regarding dietary practices of adolescents reflect the interplays of environmental and sociocultural trends in food availability, nutritional goals and preferability. A survey has indicated that the total energy intake of sweet snacks and candies has increased, while fruit juice and vegetable consumption has decreased among adolescents due to increased attainability of sweet snacks in their living environment [8]. This trend was postulated by Amos et al. [9] that availability of toffees and soft drinks on the school grounds, as well as in the participants' communities, could contribute to the high intake of soft drinks and toffees in the region.

The growing concern is that dietary habits of most societies have changed considerably during a short period of time due to great economic development, as unhealthy foods such as candies, fast foods, and soft drinks become more accessible and financially feasible than healthier options, both in schools and communities. This has fueled increased consumption of unhealthy food, and, in turn, increased undesirable health consequences [10,11].

Although healthy eating is necessary during adolescence, studies have shown that as the individual enters adolescence, their dietary habits often become unhealthier [12,13]. For young people, healthy eating is particularly important for growth and cognitive development. Eating behaviours adopted during adolescence are likely to be maintained into adulthood, underscoring the importance of encouraging healthy eating as early as possible. Guidelines recommend consumption of at least five portions of fruits and vegetables per day, reduced intake of saturated fat and salt, and increased consumption of complex carbohydrates, as these are essential for adolescent growth and vitality [11].

The global increase in prevalence of obesity in children and adolescents could be attributed to unhealthy dietary practices of adolescents [14]. Obesity in childhood is associated with medical and psychosocial complications and is likely to track into adulthood [3]. Prevention and intervention programmes for children need to start early, focusing on family and school-based initiatives and other adverse environmental factors that overwhelm behavioural and educational techniques designed to modify diet and increase participation in physical activity [3]. In light of the obesity epidemic, it is perhaps not surprising that dieting is such a prevalent behaviour among children [14], some of whom employ unhealthy dieting strategies to lose weight [1].

Eating habits acquired during adolescence continue into adulthood where the association between diet, disease morbidity, and mortality is well recognized [2,3]. Healthy eating habits contribute to the physical and emotional health and well-being of adolescents. However, psychosocial changes during adolescence (associated with a need for increased independence), environmental factors (advertising, peer pressure, spending more time away from home), and consuming greater quantities of fast foods and snacks have a combined effect on adolescents' eating patterns and food choices. These factors may put adolescents at increased risk for unhealthy eating habits, resulting in poor nutritional health [15].

In addition, dieting behaviours may have a bearing on the health status of adolescents, as there is widespread concern about excessive dieting among young people. The changes in trends of availability, accessibility and visibility of fast food, candies, cookies and soft drinks in schools and communities have been found to be responsible for the increased prevalence of obesity and cardiovascular disease in the developing world. The extent to which these environmental and sociocultural factors interlock and contribute to the changing trends of dietary choices of adolescents in Ghana is unknown. Therefore, examining dietary practices of in-school adolescents and contributory factors is essential for cost effective, culturally-sensitive policy intervention to decrease unhealthy dietary practices and increase intake and enjoyment of fruits and vegetables in our society.

## Materials and methods

A school-based cross-sectional approach was used to select study participants from the accessible population consisting of all in-school adolescents in JHS within the age of 10–15 years. Adolescents at this level of education are between 10–15 years and per this study are found in Forms 1–3 [16]. The study subjects were characterized into three groups; 10–11, 12–13, and 14–15 years. However, all of the students sampled were above 11 years, hence, analysis was done with two groups; 12–13 and 14–15 years [17]. A multistage sampling method was

adopted comprising cluster, simple random and convenience procedures to select 1,400 in-school adolescents in the JHS in the Central region of Ghana. The sample size was calculated using Cohen 'G' power with an effect size of .40, confidence level of 95% and confidence interval of .05. Yet, 94% (N = 1,311) of the questionnaires were completed, returned and used for the analysis.

## Survey instrument

A modified version of the generic Global School Health Survey [18] questionnaire was adapted to suit the context of this study. The original questionnaire contained 84 items posted on unintentional and intentional injuries, tobacco use, alcohol and other drug use, high risk sexual behaviours, dietary behaviours and physical activity among adolescents, however, the current study made use of only the dietary behaviour questions. The questionnaire consisted of 12 items that solicited responses from the participants. Six items sought socio-demographic information and six items measured respondents' dietary practices, including breakfast intake, fruit and vegetable consumption, frequency of carbonated drink consumption and frequency of sweets/toffees consumption. The internal consistency measure for the instrument yielded a reliability coefficient of Kuder-Richardson [KR20] formula 0.80. This value indicated high reliability and KR20 reliability was suitable for dichotomous variables.

## Data collection procedure

Selected schools' headmasters/mistresses were formally notified through the metropolitan, municipal/district education directors. Parental involvement could affect participants' responses, hence, their consents waived, and informed consents obtained from the participants after the heads of the schools granted permission for the study. Data collection was done during closing periods of the schools in order not to disrupt contact hours. Participants were asked to answer the questionnaires after research assistants explained the importance of honesty on the part of subjects and enhanced confidentiality on the part the researcher. Participants completed the questionnaires in their classrooms and recorded their responses directly by ticking or writing their responses to the questions.

## Data analysis

Data was coded and analyzed using the Statistical Package for Social Sciences ([SPSS] version 22.0 for Windows) after pre-screenings were initially piloted to ensure accuracy and appropriateness of the data. In order to make meaning out of the data, participants' responses to how often they eat fruits and vegetables in a week were categorized into healthy and unhealthy dietary practices. Consumption of fruits and vegetable regularly or always was described as healthy dietary practices while unhealthy dietary practices, was when consumption of fruits and vegetables was sometimes or never. Frequency counts and percentages were employed to report on dietary practices among in-school adolescents in the region and Chi-square was used to determine socio-demographic disparities in the dietary practices. Further analysis was done using binary logistic regression to establish the relative influence of the socio-demographic factors on dietary practices among in-school adolescents in the region. The outcome variable was dietary practices (unhealthy vs. healthy) and predictor variables were age, sex, socioeconomic status, parental communication (classified as difficult to talk to parents or easy to discuss issues with parents), academic performance (self-reported academic performance), geographical location and religion.

## Ethical approval

The study was approved by the Institutional Review Board, University of Cape Coast [UCCIRB/CES/2016/04] after scrutiny of the study protocol where parental consent was waived off. The permission of waiving parental consents was based on the premise that parental involvement may affect students' responses, hence, parental involvement was waived with authority from WHO that when conducting a sensitive study among minors and parental involvement could result in social desirability that could affect responses from the minors, parental autonomy can be waived. Maintenance of confidentiality and anonymity of the participants was ensured throughout the study. Teachers were also excused from the classrooms and students asked to fold the answered questionnaires without their names and place it in a box provided in front of the classroom to ensure anonymity and confidentiality. Participation in the study was voluntary without any compulsion.

## Results

Dietary practices were assessed and the responses were aggregated. It was found that 49.9% (n = 654) of in-school adolescents regularly consumed fruits and vegetables (Table 1). Also, high consumption of toffees and soft drinks among in-school adolescents was found, as 90% (n = 1183) of the participants regularly consumed toffees and 93% (n = 1220) regularly consumed soft drinks (Table 1). The result also revealed 57% (n = 749) of participants had never taken breakfast before going to school (Table 1). Early morning school attendance could be responsible for this outcome.

### Sociodemographic variations of dietary practices

There was found to be a statistically significant difference among students in terms of gender, age, socioeconomic status, parental communication, academic performance, and geographical location as shown in Table 2. The results in Table 2 showed that 54% (n = 367) of girls engaged in healthy dietary practices compared to 46% of boys (n = 287) [$\chi^2$ (df = 1) = 7.818, P = .005]. This outcome meant that girls were more likely to engage in the healthy dietary practice of eating fruits and vegetables than boys. In addition, 57% (n = 171) of in-school adolescents within the age group of 12–13 years were found to have engaged in healthy dietary practices, compared to those within the age group 14–15 years [$\chi^2$ (df = 1) = 7.876, P = .005]. This outcome reveals that unhealthy dietary practices increase with age. This outcome calls for intensification of school regulation to ban the selling of junk food, toffees and carbonated drinks on school grounds and promote the selling of healthy food, such as fruits and vegetables. In addition, a statistically significant difference was found in terms of parental communication and healthy dietary practices [$\chi^2$ (df = 1) = 5.812, P = .016]. It was observed that adolescents with easy parental communication 52% (n = 439) were more likely to consume fruits and vegetables

**Table 1. Dietary practices of in-school adolescents.**

| Dietary Practices | Regular Intake (Yes) | | Irregular Intake (No) | |
|---|---|---|---|---|
| | Freq. | % | Freq. | % |
| Fruits and vegetable | 654 | 49.9 | 657 | 50.1 |
| Carbonated drinks | 1220 | 93.1 | 91 | 6.9 |
| Sweets/Toffees | 1183 | 90.2 | 128 | 9.8 |
| Breakfast intake | 562 | 42.9 | 749 | 57.1 |

*N* = 1311.

**Table 2. Sociodemographic variations in the dietary practices.**

| Factors | Dietary Practices Unhealthy Healthy f % f % | | | | $\chi^2$ | Df | P-value |
|---|---|---|---|---|---|---|---|
| **Gender** | | | | | | | |
| Boys | 339 | 54.2 | 287 | 45.8 | 7.818 | 1 | .005 |
| Girls | 318 | 46.4 | 367 | 53.6 | | | |
| **Age** | | | | | | | |
| 12-13yrs | 129 | 43.0 | 171 | 57.0 | 7.876 | 1 | .005 |
| 14-15yrs | 528 | 52.2 | 483 | 47.8 | | | |
| **SES** | | | | | | | |
| Low | 449 | 52.0 | 414 | 48.0 | 3.698 | 1 | .054 |
| High | 208 | 46.4 | 240 | 43.6 | | | |
| **PC** | | | | | | | |
| Difficult | 258 | 54.5 | 215 | 44.5 | 5.812 | 1 | .016 |
| Easy | 399 | 47.6 | 439 | 52.4 | | | |
| **AP** | | | | | | | |
| Below Ave. | 159 | 63.3 | 92 | 36.7 | 30.682 | 2 | .001 |
| Average | 330 | 50.6 | 322 | 49.4 | | | |
| Above Ave. | 168 | 41.2 | 240 | 58.8 | | | |
| **Religion** | | | | | | | |
| Christians | 585 | 50.5 | 574 | 45.5 | .519 | 1 | .471 |
| Muslims | 72 | 47.4 | 80 | 52.6 | | | |
| **Geography Location** | | | | | | | |
| Southern | 253 | 57.4 | 188 | 42.6 | | | |
| Middle | 192 | 41.0 | 278 | 59.0 | 26.750 | 2 | .001 |
| Northern | 212 | 53.0 | 188 | 47.0 | | | |

N = 1, 311. SES = Socio-economic Status, AP = Academic Performance and PC = Parental Communication.

than adolescents with difficult parental communication. Furthermore, students with above average academic performance 59% (n = 240) were found to have engaged in healthy dietary practices than students with average and below average academic performance [$\chi^2$ (df = 2) = 30.682, P = .001]. The findings mean that in-school adolescents whose academic performances are above average know the importance of healthy dietary practices. Also, it means that knowledge about the essence of fruits and vegetables increase healthy dietary practices. Finally, differences were found in the geographical location [$\chi^2$ (df = 2) = 26.750, P = .001] such that 59% (n = 278) of in-school adolescents from middle/central zones consumed fruits and vegetables more than adolescents from the coastal and northern zones. A possible reason could be the higher availability of fruits and vegetables in the middle/central parts of the region.

Nevertheless, there were no statistically significant differences between in-school adolescents in terms of their socioeconomic status and religious affiliation in relation to dietary practices. This result means that to engage in the healthy dietary practice of consuming fruits and vegetables is not dependent on the financial status of the parents or religious affiliation.

## Socio-demographic factors predict healthy dietary practices among in-school adolescents

A binary logistic regression was conducted to find the strength of associations between socio-demographic factors and healthy dietary practices among in-school adolescents in the region.

**Table 3. Binary logistic regression of socio-demographic predictors of healthy dietary practices among in-school adolescents.**

| Predictors | N | % | B | Wald | OR | 95% CI | Sig. |
|---|---|---|---|---|---|---|---|
| **Age** | | | | | | | |
| 12-13years (ref) | 171 | 26.1 | | | | | |
| 14-15years | 483 | 73.9 | 0.243 | 3.153 | .785 | .60–1.03 | 0.076 |
| **Gender** | | | | | | | |
| Boys (ref) | 287 | 43.8 | | | | | |
| Girls | 367 | 56.2 | 0.309 | 7.30 | 1.36 | 1.1–1.70 | 0.007 |
| **Religious Affiliation** | | | | | | | |
| Christian (ref) | 574 | 87.8 | | | | | |
| Muslims | 80 | 12.2 | 0.960 | 0.288 | 1.10 | 0.77–1.56 | 0.592 |
| **Parental Communication** | | | | | | | |
| Difficult (ref) | 215 | 32.9 | | | | | |
| Easy | 439 | 67.1 | 0.210 | 3.09 | 1.23 | 0.97–1.56 | 0.078 |
| **Socioeconomic Status** | | | | | | | |
| Low (ref) | 414 | 63.3 | | | | | |
| High | 240 | 36.7 | 0.112 | .843 | 1.11 | 0.88–1.42 | 0.358 |
| **Academic Performance** | | | | | | | |
| Below average (ref) | 92 | 14.1 | | 20.52 | | | 0.001 |
| Average | 322 | 49.2 | 0.460 | 8.502 | 1.59 | 1.2–2.16 | 0.004 |
| Above average | 240 | 36.7 | 0.784 | 20.42 | 2.19 | 1.6–3.08 | 0.001 |
| **Geographical Location** | | | | | | | |
| Southern (ref) | 188 | 28.7 | | 21.64 | | | 0.001 |
| Central | 278 | 42.5 | 0.583 | 17.65 | 1.79 | | 0.001 |
| Northern | 188 | 28.7 | 0.057 | .159 | 1.06 | | 0.690 |
| Constant | | | 0.864 | 16.43 | .422 | | 0.001 |

Source: Field Survey 2017 ref = Reference group.

Odds ratios (OR) and confidence intervals (CI) were used to determine the strength of the association. The results in Table 3 showed that the whole model significantly predicted healthy dietary practices among in-school adolescents (-2LogL = 1747.5, $\chi^2$ = 69.83, P = 0.001). Over-all, socio-demographic characteristics of in-school adolescents explained 6.9% of the variance in healthy dietary practices in the region as Nagelkerke $R^2$ indicated a variance of 0.069.

Results showed statistically significant variations in the odds of eating fruits and vegetables within gender, academic performance and geographical location. The result revealed that girls were 1.4 times more likely to eat fruits and vegetables than boys (OR = 1.36, 95% CI = 1.08–1.70, P = 0.007). This means that boys are at higher risk of unhealthy dietary practices than girls. Similarly, students with above average academic performance were 2.2 times more likely to engage in healthy dietary practices than those whose performances were average and below average (OR = 2.19, 95% CI = 1.56–3.08, P = 0.001). Also, students with average academic per-formance were 1.6 times more likely to eat fruits and vegetables than those with below average performance (OR = 1.59, 95% CI = 1.16–2.16, P = 0.004). In addition, students in the central part of the region were 1.8 times more likely to engage in healthy dietary practices than those from the southern part (OR = 1.79, 95% CI = 1.37–2.35, P = 0.001). Also, adolescents from the northern part of the region were not found to consume fruits and vegetables more than those from the southern part of the region (OR = 1.06, 95% CI = 0.97–1.56, P = 0.690). However, no statistically significant variations were found in the odds of eating fruits and vegetables within age (OR = 0.79, 95% CI = 0.60–1.03, P = 0.076), religious affiliation (OR = 1.10, 95%

CI = 0.77–1.57, *p* = 0.592), parental communication (OR = 1.23, 95% CI = 0.98–1.56, *P* = 0.078) or socioeconomic status (OR = 1.12, 95% CI = 0.88–1.42, *P* = 0.358).

## Discussion

Adolescents' dietary practices play a key role in growth and development of adolescents. Absence of quality in food choices promote obesity and cardiovascular diseases in adolescents. Overall dietary behaviour during the period is anchored on monotony of food choices from energy dense foods and sweets. Lack of quality in food choices among this group is the driving force of obesity burden globally [19]. Healthy choices reduce the dual burden of overeating and under eating during this period. The study found that close to half of the adolescents in the Central region of Ghana practiced healthy dietary of fruits and vegetables consumption. The overall prevalence of healthy dietary practices was 49.9%. This means that half of the adolescent population in the region engage in unhealthy dietary practices. Unhealthy dietary practices is associated with increased risks for obesity and cardiovascular diseases. The finding is consistent with previous findings that healthy dietary practices regarding fruit and vegetable consumption have hovered around 40–50% among adolescents globally [19–21]. Even though half of the participants did not consume fruits and vegetables regularly, quite a substantial number consumed carbonated drinks and toffees regularly. This means that most adolescents are engaging in unhealthy dietary practices that could be detrimental to their health in the future. As most of the in-school adolescents are not consuming the recommended servings of fruits and vegetables, majority of them could be deprived of the protective benefits of fiber in their diets, making them at higher risk for obesity and cardiovascular diseases [10]. The study also found high level of soda and sweet consumption among in-school adolescents in the region. The increased consumption of sweets and soft drinks could be attributed to high proliferation of carbonated drinks and sweet in the schools and community due globalization. This is because the environment is major contributory factor in food choices. The high level of soda and sweets consumption could subsequently lead to increase prevalence of obesity and cardiovascular diseases among adolescents. This means that school policies should be promulgated against the sale of carbonated drinks and sweets in the schools. These bylaws should be enforced by the schools to ensure healthy variety food choices in the schools.

Adolescents' concentration in the learning process in the classroom settings with an enhanced attention span is influenced by their nutritional states. Eating before school is essential for good health and vitality. This means that healthy breakfast is critical in enhancing concentration, attention process and healthy behaviours in the school environment. In the study, a low intake of breakfast (43%) was found among the students in the region where most in-school adolescents never ate before school. The low intake of breakfast before early morning learning means students may lack essential nutrients that support brain function, concentration and attention during early morning lessons. A plausible reason for low breakfast consumption among students in the region might be due to the early reporting time to school each day and the distance students commute to arrive at school. This finding is inconsistent with the Intiful and Lartey [22] research reports, where they found high breakfast consumption among senior high school adolescents. This incongruence could be due to differences in the sampled population while the previous study sampled boarding school students where breakfast is compulsory, the current study used non-boarding students, JHS. Another reason for low breakfast consumption in the region may be poverty [22]. Breakfast is essential for students' concentration in class. Students who skip breakfast lack essential nutrients for the brain, as a good breakfast gives students physical energy and boosts their mental capacity [23]. With

this, it is essential for parents to ensure that their children eat a healthy breakfast every day to improve their health status, concentration in class and academic performances.

## Socio-demographic variations in dietary intake

Health inequities is a major issue observed in the dietary practices among adolescents. This study observed disparities in terms of gender, age, socioeconomic status, parental communication, academic performance, and geographical location in relation to healthy dietary practices. These disparities are critical in terms proffering appropriate interventions to enhance healthy dietary practices and reduce prevalence of adolescent obesity. The finding that girls had healthier dietary practices than boys corroborates with Ranjit et al. [24] and Nilsen et al. [25] that the healthy dietary practice of fruit and vegetable intake is often higher among girls compared to boys. This observation prompts the need for increased physical activity in school among boys to increase better glucose metabolism and reduce the effect of unhealthy dietary practices on their body weight. Increased education on healthy dietary practices is also necessary.

The finding that the younger age group had healthier dietary practices than the older age group supports the findings of Ranjit et al. [24] and Pearson et al. [26] that younger adolescents engaged in healthier dietary practices than older ones. Harnack et al. [27] explained that soft drink and fast food consumption increased with age in adolescents in the United States. Similarly, Bere et al. [28] study in Norway posited that fast food consumption and the frequency of visiting fast food restaurants was higher among students in upper classes (ages 15–18) than students in lower classes (ages 12–14).

In terms of academic performance, the finding that students with above average academic performance had healthier dietary practices than those with average or below average performance is consistent with Falkner et al. [29] who found that unhealthy dietary behaviors are always found among students who have had to repeat their grade. The adolescents in the Central Region did not differ in terms of their socioeconomic status and dietary practices. This finding contrasts with the finding of Henningsen [30] that adolescents with low socio-economic position have a higher consumption of soft drinks and fast food than those with high socio-economic position. Also in contrast with this study, but similar to Fahlman et al. [31] also found that those with high socioeconomic position engaged in healthier dietary practices. Variations exist in adolescents dietary practices, adolescents from less affluent homes generally report unhealthy eating practices, however, affluence homes also influence consumption soda and soft drinks.

## Predictors of dietary intake

Gender was a significant predictor of healthy dietary practices among in-school adolescents in the Central region with boys at more risk of engaging in unhealthy dietary practices. Unhealthy dietary practices may contribute to developing unhealthy weights and increasing the burden of obesity and CVDs among boys. This outcome corroborates with previous research findings [32]. These studies reported that girls were more likely to engage in the healthy dietary practice of eating fruits and vegetables than boys. The plausible explanation to these similarities could be that girls prefer to eat at home compared to boys. Although Layade and Adeoye [32] also found gender as a predictor of healthy dietary practices, they posited that males were more likely to consume fruits and vegetables than females. The possible reason for this difference in the finding could be because the current study sampled junior high school students while the previous study sampled tertiary students. The outcome also means that boys would lack essential nutrients from fruits and vegetables and suffer related consequences, resulting in the need to promote fruit and vegetable consumption among boys.

Furthermore, academic performance of in-school adolescents also predicted healthy dietary practices in the region, as adolescents with above average academic performance were more likely to eat fruits and vegetables than those with average or below average academic performance. This finding is consistent with previous research findings of Doku et al. [33] that academic performance of adolescents was a significant predictor of healthy dietary practices among adolescents, with students having above average academic performance being more likely to consume fruits and vegetables. The confirmatory reason for this outcome could be that students with high academic performance are aware of the importance of fruits and vegetables in their diet, and this could be responsible for high consumption of fruits and vegetables among this group [34]. Another plausible reason for the similarities in the findings could be that students with high academic performance eat fruits and vegetables because availability and purchasing ability of the students. Therefore, there is a need for education on the importance of fruits and vegetables among students with average and below average academic performance. Also, an effective intervention of implementing a healthy canteen intervention programme in the schools as recommended by Mohammadi et al [3] could be the surest of to improve students' dietary intake.

Furthermore, this study also found that in-school adolescents in the southern part of the region were at higher risk of not eating fruits and vegetables than adolescents from the central part of the region. This could be as result of unavailability of fruits and vegetables in the southern part of Ghana and if available they are very expensive. This finding is in support of the previous research findings by Layade and Adeoye [32] and Othman et al. [35] on fruit and vegetable consumption by students in Nigeria, where they described geographical area and availability of fruits and vegetables in the community as a predictor of fruit and vegetable consumption among adolescents. This explains that availability and access to fruits and vegetables in a locality determine students' consumption. Similarly, the study affirms the research report of Currie et al. [36] for HBSC, that the engagement in health behaviours patterned geographical differences, hence the explanation that certain health behaviours depicted certain patterns across geographical areas. This meant that health outcomes showed the pattern of life, culture, traditions and weather conditions of the geographical zones of people. Also, the outcome could be due to no or limited availability of fruits and vegetables on school campuses in the southern part of the region, contributing to inhibited intake, hence, there is a need to encourage the consumption of fruits and vegetables in schools. Nutritional needs during adolescence are increased due to the increased growth rate and changes in body composition associated with puberty [6]. However, poor dietary practices among school adolescents could contribute to obesity. Fruit and vegetable consumption should be promoted in our schools, especially among boys and adolescents living in the northern and southern parts of the region.

## Limitations

The cross-sectional approach adopted could limit the generalizability of the predictors of fruits and vegetable consumption among in-school adolescents. Although the utilization of self-recall on the dietary practices of adolescents is useful in determining intake over a long period of time, it lacks detailed information on specific foods and beverages and memory biases and other social desirability issues inherent in the study may lead to over or under reporting of dietary practices.

## Conclusions

The findings from this school-based survey have revealed that there is low fruit and vegetable intake and high consumption of sweets and carbonated drinks in Ghana among adolescents.

These food preferences have increased the propensity of influencing the burdens of adolescent obesity and cardiovascular diseases in the region. There is a need to increase availability of fruits and vegetables in schools to compliment the low intake that may occur at home. Also, nutritional education may be a suitable focus for the improvement of dietary behaviors among adolescents in the country. Fruit and vegetable consumption was associated with gender, academic performance and geographical location, and these may be a reflection that knowledge and availability are important factors influencing dietary practices and food choices. School health policy interventions aimed at improving nutritional status among adolescents and enhanced fruit and vegetable consumption in the country should consider the potential benefit of increasing the availability of fruits and vegetables in schools, while reducing access to sweets and soft drinks in schools and communities.

## Supporting information

**S1 File.**
(SAV)

## Acknowledgments

Special thanks to the heads of Basic Schools in the Central Region and the students for making this research possible. The author wish to thank Elyssa Shoup of Seattle, Washington State-USA for proof reading and editing of this manuscript.

## Author Contributions

**Conceptualization:** Thomas Hormenu.

**Data curation:** Thomas Hormenu.

**Formal analysis:** Thomas Hormenu.

**Funding acquisition:** Thomas Hormenu.

**Investigation:** Thomas Hormenu.

**Methodology:** Thomas Hormenu.

**Project administration:** Thomas Hormenu.

**Resources:** Thomas Hormenu.

**Supervision:** Thomas Hormenu.

**Validation:** Thomas Hormenu.

**Visualization:** Thomas Hormenu.

**Writing – original draft:** Thomas Hormenu.

**Writing – review & editing:** Thomas Hormenu.

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
