## [Decision Letter · Decision Letter 0]

6 Dec 2021

PONE-D-21-08718

Dietary Intake and Associated Factors among In-school Adolescents in Ghana

PLOS ONE

Dear Dr. Hormenu,

Thank you for submitting your manuscript to PLOS ONE. After careful consideration, we feel that it has merit but does not fully meet PLOS ONE’s publication criteria as it currently stands. Therefore, we invite you to submit a revised version of the manuscript that addresses the points raised during the review process.

Please address all reviewer comments. Please note that you are not requested to cite the references listed by the reviewers; you may consider whether they are relevant or not and, in your revisions, choose whether to cite them or not.

We look forward to receiving your revised manuscript.

Yours sincerely,

Yann Benetreau, Ph.D.

Senior Editor, *PLOS ONE*

Journal Requirements:

"No funder has a role in the study design, data collection, analysis and decision to publish this manuscript"

7. We note you have included a table to which you do not refer in the text of your manuscript. Please ensure that you refer to Table 2 in your text; if accepted, production will need this reference to link the reader to the Table.

Reviewers' comments:

Reviewer's Responses to Questions

**Comments to the Author**

1. Is the manuscript technically sound, and do the data support the conclusions?

Reviewer #1: No

Reviewer #2: Yes

2. Has the statistical analysis been performed appropriately and rigorously? 

Reviewer #1: Yes

Reviewer #2: Yes

3. Have the authors made all data underlying the findings in their manuscript fully available?

Reviewer #1: Yes

Reviewer #2: Yes

4. Is the manuscript presented in an intelligible fashion and written in standard English?

Reviewer #1: Yes

Reviewer #2: Yes

5. Review Comments to the Author

Reviewer #1: Thank you for providing insight into the dietary intake of adolescents in Ghana. I suggest that you give more detail to the changes you made to the Global School Health Survey, reference 21 (2012 fact sheet). Only item on the fact sheet is used in your study. There is no reference to socio-economic factors in your conclusion, but you have used as a keyword. Good luck with the revisions.

Reviewer #2: 1. Introduction should be shorter

2. Proofreading and paraphrasing by an expert are necessary. Several sentences have to be revised.

3. Number of references are not sufficient. Some references are too old. Please replace them with the updated references

The following references for reading are suggested.

-Mohammadi S, Jalaludin MY, Su TT, Dahlui M, Mohamed MN, Majid HA. Dietary and physical activity patterns related to cardio-metabolic health among Malaysian adolescents: a systematic review. BMC public health. 2019 Dec;19(1):1-9.

-Mohammadi S, Jalaludin MY, Su TT, Dahlui M, Azmi Mohamed MN, Abdul Majid H. Determinants of diet and physical activity in Malaysian adolescents: a systematic review. International journal of environmental research and public health. 2019 Jan;16(4):603.

-Mohammadi S, Su TT, Papadaki A, Jalaludin MY, Dahlui M, Mohamed MN, Jago R, Toumpakari Z, Johnson L, Majid HA. Perceptions of eating practices and physical activity among Malaysian adolescents in secondary schools: a qualitative study with multi-stakeholders. Public health nutrition. 2021 Jun;24(8):2273-85.

4. Methods: Please add statement for the information sheet and consents obtained from parents of students.

5. Discussion should be longer with more references and comparisons with other studies

6. Conclusion section should be revised.

6. PLOS authors have the option to publish the peer review history of their article (what does this mean?). If published, this will include your full peer review and any attached files.

Reviewer #1: No

Reviewer #2: **Yes: **Dr. Shooka Mohammadi

---

## [Author Response · Author response to Decision Letter 0]

10 Feb 2022

The author received no specific funding for this research

b) State what role the funders took in the study. If the funders had no role in your study, please state: “The funders had no role in study design, data collection and analysis, decision to publish, or preparation of the manuscript.” No funding role in data collection, processing and analysis.

c) If any authors received a salary from any of your funders, please state which authors and which funders. The author received no specific funding for this work.”

5. We note that you have indicated that data from this study are available upon request. PLOS only allows data to be available upon request if there are legal or ethical restrictions on sharing data publicly. For more information on unacceptable data access restrictions, please see http://journals.plos.org/plosone/s/data-availability#loc-unacceptable-data-access-restrictions. The data will be made available for the replication of the study. Thank you.

a) If there are ethical or legal restrictions on sharing a de-identified data set, please explain them in detail (e.g., data contain potentially sensitive information, data are owned by a third-party organization, etc.) and who has imposed them (e.g., an ethics committee). Please also provide contact information for a data access committee, ethics committee, or other institutional body to which data requests may be sent. There is no ethical restriction on the data.

We will update your Data Availability statement on your behalf to reflect the information you provide. Data will be made available.

6. We note that you have stated that you will provide repository information for your data at acceptance. Should your manuscript be accepted for publication, we will hold it until you provide the relevant accession numbers or DOIs necessary to access your data. If you wish to make changes to your Data Availability statement, please describe these changes in your cover letter and we will update your Data Availability statement to reflect the information you provide. It will be updated accordingly.

7. We note you have included a table to which you do not refer in the text of your manuscript. Please ensure that you refer to Table 2 in your text; if accepted, production will need this reference to link the reader to the Table. Reference has been to Table 2 in the manuscript at page 5 that readers can accordingly have reference to this table. 

8. Please review your reference list to ensure that it is complete and correct. If you have cited papers that have been retracted, please include the rationale for doing so in the manuscript text, or remove these references and replace them with relevant current references. Any changes to the reference list should be mentioned in the rebuttal letter that accompanies your revised manuscript. If you need to cite a retracted article, indicate the article’s retracted status in the References list and also include a citation and full reference for the retraction notice. The references has been updated.

Reviewers' comments:

Reviewer's Responses to Questions

Comments to the Author

1. Is the manuscript technically sound, and do the data support the conclusions?

Reviewer #1: No

Reviewer #2: Yes

2. Has the statistical analysis been performed appropriately and rigorously?

Reviewer #1: Yes

Reviewer #2: Yes

3. Have the authors made all data underlying the findings in their manuscript fully available?

Reviewer #1: Yes

Reviewer #2: Yes

4. Is the manuscript presented in an intelligible fashion and written in standard English?

Reviewer #1: Yes

Reviewer #2: Yes

5. Review Comments to the Author

Reviewer #1: Thank you for providing insight into the dietary intake of adolescents in Ghana. I suggest that you give more detail to the changes you made to the Global School Health Survey, reference 21 (2012 fact sheet). Only item on the fact sheet is used in your study. There is no reference to socio-economic factors in your conclusion, but you have used as a keyword. Good luck with the revisions. I thank reviewer 1 for these suggestions, the revision has been according made in the manuscript. The original questionnaire contained 84 items posted on unintentional and intentional injuries, tobacco use, alcohol and other drug use, high risk sexual behaviours, dietary behaviours and physical activity among adolescents, however, the current study made use of only the dietary behaviour questions.

Reviewer #2: 1. Introduction should be shorter. I thank the Reviewer 2 for this suggestion, but I think the introduction set the rationales underpinning the research, hence, the author elaborated on adolescent life and behaviour formation as well as fruit and vegetable consumption as the foundation for the study. I think, this aspect of the manuscript espoused the basis for the study. 

2. Proofreading and paraphrasing by an expert are necessary. Several sentences have to be revised. Thank you, some modifications have been made to improve the quality of the manuscript. 

3. Number of references are not sufficient. Some references are too old. Please replace them with the updated references

The following references for reading are suggested.

-Mohammadi S, Jalaludin MY, Su TT, Dahlui M, Mohamed MN, Majid HA. Dietary and physical activity patterns related to cardio-metabolic health among Malaysian adolescents: a systematic review. BMC public health. 2019 Dec;19(1):1-9.

-Mohammadi S, Jalaludin MY, Su TT, Dahlui M, Azmi Mohamed MN, Abdul Majid H. Determinants of diet and physical activity in Malaysian adolescents: a systematic review. International journal of environmental research and public health. 2019 Jan;16(4):603.

-Mohammadi S, Su TT, Papadaki A, Jalaludin MY, Dahlui M, Mohamed MN, Jago R, Toumpakari Z, Johnson L, Majid HA. Perceptions of eating practices and physical activity among Malaysian adolescents in secondary schools: a qualitative study with multi-stakeholders. Public health nutrition. 2021 Jun;24(8):2273-85. Thank you for these, suggestion, these articles have been perused and some referenced in the work, 

4. Methods: Please add statement for the information sheet and consents obtained from parents of students. I thank the reviewer 2 for this suggestion. The suggestion has been incorporated into the manuscript to enhance the quality of work. Research Study approval was obtained from the school and students signed the informed consent even though they are minor. This is because parental involvement could affect their responses, therefore, parental consents were waived. 

5. Discussion should be longer with more references and comparisons with other studies. Further discussion was done, and additional references incorporated.

6. Conclusion section should be revised. The discrepancy detected in the conclusion has been worked on. 

6. PLOS authors have the option to publish the peer review history of their article (what does this mean?). If published, this will include your full peer review and any attached files.

Do you want your identity to be public for this peer review? For information about this choice, including consent withdrawal, please see our Privacy Policy.

Reviewer #1: No

Reviewer #2: Yes: Dr. Shooka Mohammadi

EDITOR’S COMMENTS

Page 7 ...increased frequency in consumption of......? I thank the reviewer for this correction, your input has been incorporated into the manuscript. 

Page 8: See comment in conclusion of article. The necessary correction has been done

Page 8: Any statistics overweight and obesity in Ghana? Thank you for this question, there are statistics of overweight and obesity in Ghana. Some statistics have been provided in the manuscript. 

Page 8: I suggest you only refer to fast food and not to brand name. Thank you for this suggestion, the correction has been done. 

Page 9: Only 1 of the items found in reference 21. Is there another reference you can refer to? The items, here mean questions that respondents responded to. Hence, the research used 11 questions originator of the questionnaire has been referenced according

Page 10: Did schools need to give approval for study to be performed? Was consent and/or assent obtained? Yes, approval was obtained from the school and students signed the informed consent even though they are minor. This is because parental involvement could affect their responses, therefore, parental consents were waived. 

Page 10: Who did the analysis? The author did the analysis and that has been indicated accordingly.

Page 10: How was confidentiality and anonymity ensured? Confidentiality and anonymity ensured when students were asked not to write their names on the questionnaire and also after answering the questions, they were to fold the questionnaire and place it in box provided in front of the class. 

Page 11: Explain how difficult parental communication and below average performance were identified. Thank you for this suggestion. These have been incorporated; parental communication (classified as difficult to talk parents or easy to discuss issues with parents), academic performance (self-reported academic performance),

Page 11: Under material and methods, it was stated that children 10-15 years were included. Care to explain 15+ years? Thank you for this great observation. It is 15 years and not 15+ years. That correction has been made in the manuscript. 

Page 11: Are abbreviations necessary needed? SES not explained after table. The explanations to the abbreviations have been indicated as footnotes for the tables. 

Page 12: Explain what ref mean. The ref means reference group and that has been indicated as footnotes to the table. 

Page 13: Better word for school start time? This is the time schools open each day in Ghana, but that sentence has been accordingly rephrased. 

Page 13: SHS: Senior high school? Yes 

Page 13: Reference for this statement? The reference has been provided

Page 14: What about education to improve dietary habits? Yes, education to improve dietary practices is also necessary. The addition has been made in the manuscript. I thank the reviewer for this suggestion. 

Page 14: Where was Bere et al. study done? Norway and that has been indicated in the manuscript. 

Page 14: reason for this statement? Because of the population sample, differences is more likely to occur in terms of gender. The reason for gender prediction is due to differences in the sampled population. The current study sampled junior high school students while the previous study sampled senior high school students. 

Page 15: How can you say that academic performance is related to income…Thank you for this question, the academic performance was not linked to income, I was suggesting a plausible association between participants with high academic performance ability to eat fruit and vegetables could be due to ability to buy the foods and availability. The statement has been rephrased in the manuscript. 

Page 15: Is locality used in Ghana? What about area?......locality and area have been used interchangeably in Ghana but your suggested adhere to and the word changed accordingly. 

Page 15: Socio-economic factors a key word, but nothing on that in conclusion. Conclusion has been rephrased to show the socioeconomic factors associated with fruit and vegetable consumption. 

Page 16: Was data analysis done by author or are there others that should be mentioned? The author did the analysis himself.

Again, I thank the Reviewers for the thoughtful comments, queries and suggestions.

---

## [Editor Report · Decision Letter 1]

18 Feb 2022

PONE-D-21-08718R1Dietary Intake and Associated Factors among In-school Adolescents in GhanaPLOS ONE

Dear Dr. Hormenu,,

Thank you for submitting your manuscript to PLOS ONE. After careful consideration, we feel that it has merit but does not fully meet PLOS ONE’s publication criteria as it currently stands. Therefore, we invite you to submit a revised version of the manuscript that addresses the points raised during the review process.

Please note that due to the extent of the requested revisions, a detailed response to all of the editorial’ comments will be needed and the new version will go out again for a re-review. We may decide to request additional revisions, or even reject, the manuscript at that point, if we consider that the concerns have not be adequately addressed.

[1] Kindly be informed that the  In-text citations are incorrect. for example “during adolescence and adulthood [3, 42]”.

[2] Paraphrasing should be done by a native proofreader.

[3] Some references are too old. Please replace them with the updated references.

[4] Discussion should be better.

We look forward to receiving your revised manuscript.

Kind regards,

Dr.Shooka Mohammadi, Ph.D

Academic Editor

PLOS ONE
---

## [Author Response · Author response to Decision Letter 1]

20 Apr 2022

The reviewers comments have been incorporated in the work to improve its viewer readability. The old citations have been revised.

---

## [Editor Report · Decision Letter 2]

28 Apr 2022

Dietary Intake and Its Associated Factors among In-school Adolescents in Ghana

PONE-D-21-08718R2

Dear Dr. Thomas Hormenu

We’re pleased to inform you that your manuscript has been judged scientifically suitable for publication and will be formally accepted for publication once it meets all outstanding technical requirements.

Kind regards,

Dr. Shooka Mohammadi, Ph.D

Guest Editor

PLOS ONE
---

## [Editor Report · Acceptance letter]

4 May 2022

PONE-D-21-08718R2 

Dietary Intake and Its Associated Factors among In-school Adolescents in Ghana 

Dear Dr. Hormenu:

I'm pleased to inform you that your manuscript has been deemed suitable for publication in PLOS ONE. Congratulations! Your manuscript is now with our production department. 

Kind regards, 

on behalf of

Dr. Shooka Mohammadi 

Guest Editor

PLOS ONE